# Ventilation Modulation and Nanoparticle Deposition in Respiratory and Olfactory Regions of Rabbit Nose

**DOI:** 10.3390/ani9121107

**Published:** 2019-12-09

**Authors:** Jinxiang Xi, Mohamed Talaat, Xiuhua Si, Haibo Dong, Ramesh Donepudi, Senthil Kabilan, Richard Corley

**Affiliations:** 1Department of Biomedical Engineering, University of Massachusetts, Lowell, MA 01854, USA; Mohamed_Talaat@uml.edu; 2Department of Aerospace, Industrial, and Mechanical Engineering, California Baptist University, Riverside, CA 91752, USA; asi@calbaptist.edu; 3Department of Mechanical and Aerospace Engineering, University of Virginia, Charlottesville, VA 22903, USA; hd6q@virginia.edu; 4Sleep and Neurodiagnostic Center, Lowell General Hospital, Lowell, MA 01854, USA; drdonepudi@gmail.com; 5EMD Serono, Billerica, MA 01821, USA; Kabilansenthil@gmail.com; 6Greek Creek Toxicokinetics Consulting, LLC, Boise, ID 83701, USA; rcorley.gctc@gmail.com

**Keywords:** New Zealand white rabbit, sniffing, ventilation, nanoparticle deposition, olfaction, ventilation modulation, maxilloturbinate, ethmoturbinate

## Abstract

**Simple Summary:**

The rabbit nose has a very complex structure. How this structure accomplishes functions such as olfaction and the filtration of nanoparticles is not well understood. Moreover, rabbits often sniff; how this sniffing helps them smell better is still a myth. In this study, we examined the anatomy of a New Zealand white (NZW) rabbit nose, especially the spiral-shaped vestibule. Based on its high flow resistance, we hypothesized that the vestibule plays a pivot role in regulating flow ventilation to the olfactory region through sniffing. We observed that this expanding vestibule increased the nanoparticle deposition in the olfactory region, and that particles depositing in the olfactory region only came from a specific area of the nostril.

**Abstract:**

The rabbit nose’s ability to filter out inhaled agents is directly related to its defense to infectious diseases. The knowledge of the rabbit nose anatomy is essential to appreciate its functions in ventilation regulation, aerosol filtration and olfaction. The objective of this study is to numerically simulate the inhalation and deposition of nanoparticles in a New Zealand white (NZW) rabbit nose model with an emphasis on the structure–function relation under normal and sniffing conditions. To simulate the sniffing scenario, the original nose model was modified to generate new models with enlarged nostrils or vestibules based on video images of a rabbit sniffing. Ventilations into the maxilloturbinate and olfactory region were quantified with varying nostril openings, and deposition rates of inhaled aerosols ranging from 0.5 nm to 1000 nm were characterized on the total, sub-regional and local basis. Results showed that particles which deposited in the olfactory region came from a specific area in the nostril. The spiral vestibule played an essential role in regulating flow resistance and flow partition into different parts of the nose. Increased olfactory doses were persistently predicted in models with expanded nostrils or vestibule. Particles in the range of 5–50 nm are more sensitive to the geometry variation than other nanoparticles. It was also observed that exhaled aerosols occupy only the central region of the nostril, which minimized the mixing with the aerosols close to the nostril wall, and potentially allowed the undisruptive sampling of odorants. The results of this study shed new light on the ventilation regulation and inhalation dosimetry in the rabbit nose, which can be further implemented to studies of infectious diseases and immunology in rabbits.

## 1. Introduction

Rabbits (*Oryctolagus cuniculus*) have been used as surrogate models for humans in inhalation experiments, infectious disease studies and immunology tests. The origin that causes the responses in these studies can primarily be attributed to the irritants that deposit in the respiratory tract [1,2,3,4]. However, there is a lack of understanding in light of the behavior and fate of inhaled agents (viruses or bio-aerosols) in the respiratory tract of rabbits. Moreover, local dosimetry is a more relevant factor to assess outcomes than the total and sub-regional deposition rates [5,6,7]. To reliably predict both the total and local deposition rates becomes the critical first step to quantify the dose-response relationship, which further requires the detailed knowledge of the nasal physiology and flow dynamics within it [8,9,10].

Several studies have attempted to quantify the dosimetry and outcomes of inhaled aerosols in rabbits. Risk estimates of exposures to anthrax in rabbits were often selected to develop risk models due to their lower cost and easy-to-handle features relative to nonhuman primates [11,12,13,14]. Raabe et al. [15] measured the regional deposition after 45 min exposure of monodisperse micrometer aerosols in various kinds of small laboratory animals, including New Zealand white (NZW) rabbits. The subsequent radio-assay of respiratory tissues showed elevated filtration of 3 µm particles in the nasal-pharyngeal region, with over 90% nasal-pharyngeal doses and nearly zero pulmonary doses for aerosols larger than 5 µm, By contrast, a much lower nasal-pharyngeal deposition was measured for submicron particles [15]. Asgharian [16] developed an NZW rabbit mathematical dosimetry model to study inhaled anthrax; however, this model was limited to fine and coarse particles due to a lack of ultrafine deposition data, which is dictated by Brownian diffusion. Kabilan et al. [17] numerically studied the deposition of inhaled anthrax, whose deposition was highly dependent upon the local flow characteristics. More recently, Hess et al. [18] put forward an integrated computational–experimental approach by using in vitro data in rabbits to develop a physiologically-based biokinetic (PBBK) model, which can be combined to existent dosimetry models to consider species-specific variability. However, no data on ultrafine particle deposition in rabbits has been reported up to date, either experimentally or computationally [16].

Inhalation dosimetry of nanoparticles is critical in studying infectious diseases, toxicology and immunology, which can include viruses, chemical vapors and bacteria. Viruses are the smallest agents of infectious diseases, whose diameters vary from 20 nm to 400 nm [19]. The largest viruses, however, can be 500 nm in diameter and 700–1000 nm in length [19]. Chemical vapors can behave like ultrafine particles in the range of 0.5–10 nm, depending on the molecular size and degree of molecular binding [20]. Most bacteria range from 0.2–2.0 µm in diameter, while some bacteria can be 20 µm in length [21].

The functions of the rabbit nose are intimately associated with its unique structure. It has been suggested by Negus that the level of the nose structural complexity can be a good index of the olfaction acuity [22]. For instance, all macrosmatic species (mice, dogs, rats, rabbits, deers, marmosets) possess an elegant, intricate nasal morphology [23,24,25,26,27,28,29,30,31,32,33,34], while such complexity is missing in microsmatic species such as monkeys [26,35,36] and human beings [32,37,38,39,40,41,42]. In most parts of their life, rabbits rely upon their exquisite olfactory system, including detecting danger, finding food, attracting mates, identifying territories. It has been shown that rabbits can sense predictors miles away, or smell food below the ground [43]. Even newborn rabbits have superior olfaction, which guides them even with closed eyes to their mothers’ nipples [44]. Rabbits often sniff or wiggle their noses, which is believed to assist them to capture or distinguish tiny traces of pheromones or chemical molecules. On the other hand, the olfactory region itself can be an important site susceptible to infectious diseases and toxic damage. Considering that the olfactory nerves are not regenerative, damages in the olfactory mucosa can be permeant [45]. A loss, or compromised, sense of smell can severely affect the quality of life for humans and the survival of rabbits.

The objective of this study is to numerically simulate the inhalation and deposition of nanoparticles in a New Zealand white (NZW) rabbit nose model with an emphasis on olfaction under normal and sniffing conditions. Specifically, we aim to (1) develop a computational model of the NZW rabbit nasal airway that can quantify particles on the total, sub-regional and local basis, (2) generate sniffing-representative models by varying the nostril opening based on images, (3) quantify the ventilation toward maxilloturbinate and ethmoturbinate with varying nostril openings, and (4) quantify the inhalation dosimetry of nanoparticles on the basis of total, sub-regional and local deposition rates under different breathing conditions. New insights will be gained from this study on the ventilation regulation and inhalation dosimetry in the rabbit nose, which can be further applied to studies of infectious diseases and immunology in rabbits. The results of this study may also have implications for the design of scent detectors or electronic noses.

## 2. Materials and Methods 

### 2.1. Study Design

To study the effects of aerosol diameters, 18 particle sizes will be considered, which are: 0.5, 1, 2, 3, 5, 10, 20, 50, 100, 200, 300, 400, 500, 600, 700, 800, 900 and 1000 nm. Special attention will be paid on the particle size of 20, 200 and 500 nm, based on the rationale that 10-nm particles represent the smallest virus, 200-nm particles represent a large virus, while 500-nm particles represent the largest virus. Moreover, 200 nm particles are most commonly used in nanomedicines [46,47] and 500 nm particles have been found to deposit the least in human and animal airways due to their low levels in both their diffusivity and inertia [48].

To study the effects of geometry variations on nanoparticle dosimetry, four new nasal models were generated by progressively deforming the nostril base of the original nose model, two by expanding the base and the other two by shrinking the base. The magnitudes of the nostril deformation were based on high-speed camera images of rabbit sniffing.

To study the flow partition, as well as the deposition distribution, in different parts of the nose, the airway model will be separated into left and right, each of which is further divided into the respiratory zone (mainly maxilloturbinate), olfactory region (ethmoturbinate) and maxillary sinus (lateral recess). Details of the model development, flow-particle transport simulations, as well as subsequent statistical analysis, will be presented in the following sections.

### 2.2. Image-Based NZW Rabbit Model (Control Case)

An anatomically accurate rabbit nasal airway geometry based on high-resolution magnetic resonance imaging (MRI) scans that were previously reported by Corley et al. was measured in this study (Figure 1a) [25]. The original data acquisition had been approved by the Institutional Animal Use Committee at Pacific Northwest National Laboratory and was not needed in this study. Figure 1 shows the image-based nose model superimposed on an NZW rabbit, with the slit-shaped nostrils and two cross-section views. Slice 1 is cut within the maxilloturbinate and Slice 2 the ethmoturbinate, or olfactory region. Corresponding MRI images are shown in Figure 1b. Both slices exhibit high complexity in morphology, with Slice 1 resembling the coral reefs and Slice 2 looking like thin scrolls. The reconstructed three-dimensional (3D) nose model is shown in Figure 1c as both the 3D-printed physical model and the computational model in different views. To quantify the airflow ventilation and particle deposition in different regions, the nose was separated into three parts: the maxilloturbinate region (gray), the olfactory region (pink) and the sinus (light blue). Considering the olfactory region (pink), even though it is closely packed with the sinus, they are indeed separate organs, with a small ostium connecting these two. Airflow and particles enter the olfactory region through a thin channel, as illustrated by the blue line (opening to OR) in Figure 1c. Obviously, the dorsal meatus is the main avenue to the olfactory region. It is noted that both the maxillo- and ethmoturbinate are internal structures within the nasal cavity. It is difficult to image the 3D structures of these conchae with conventional transparent views. To better understand these internal structures, the left nasal passage has been dissembled to reveal both the maxilloturbinate and ethmoturbinate (Figure 1d).

In the second panel of Figure 1d, the structure in green is the maxilloturbinate tissue, which is enshrouded tightly by the external walls and forms three different air passages (the first panel in Figure 1d): The dorsal meatus that leads to the olfactory region, the maxilloturbinate that leads to the trachea and the ventral meatus that leads to the vomeronasal organ (Figure 1c, red circle at the bottom of the 3D-printed model). Correspondingly, from the nostril slit the vestibule ramifies into three channels, where each has a spiral curvature and leads to the dorsal meatus (blue), maxilloturbinate (red) and ventral meatus, respectively. The rightmost panel displays the left nose with both external maxilloturbinate walls removed, as well as the transparent view of the olfactory region (i.e., ethmoturbinate).

### 2.3. Generation of Deformed Models with Expanded Nostril and Vestibule 

The original nose model (referred to as control hereinafter) was further deformed based on the observations of high-speed video images of rabbit sniffing (Figure 2a). Both the nostril and the vestibule change their shapes during sniffing. To simulate this change, the widths at the middle of the nostril slit were measured and the maximal width variation was determined. The left nostril was then progressively enlarged to generate three new models N1, N2 and N3 using HyperMorph (Troy, MI, USA), with N3 representing the geometry with the maximal nostril widening. In doing so, an enclosing box was built around the left nostril and the vestibule, and the nodes on the enclosing box were moved to change shape, which further changed the geometry in it (Figure 2b). More details of the usage of HyperMorph can be found in Xi et al. [49,50,51]. Figure 2c compares the axial contours of the nostril and vestibule between the control (black) and N3 (blue), while the nostril area and vestibule volume are compared in Figure 2d between the control and N1–N3. Considering that sniffing will also change the vestibule and even the nasal valve, a new deformed model was developed that aimed to expand the vestibule without significantly varying the nostril area. 

The resultant model is displayed in Figure 2c in green (i.e., vestibule model) in comparison to the control and N3. 

As shown in Figure 2d, the model with enlarged vestibule has a similar area to that of the control, but has the largest volume among all models considered.

### 2.4. Numercial Methods 

The rabbit inhalation flow rate is around 0.68 L/min at normal breathing conditions [52,53] and increases to 1.91 L/min during sniffing (a factor of 2.81 higher) [28]. To investigate the impact of flow rate on ventilation and particle deposition, four speeds (i.e., 0.68 L/min, 1.36 L/min, 1.91 L/min, 2.72 L/min) were included. Considering that airflow can enter the nostril nonuniformly, pressure inlet (zero) and outlet (vacuum) boundary conditions were adopted. A rigid and nonslip condition was assumed for the airway surface. The respiratory airflow was assumed to be isothermal, incompressible with steady inhalations. A low-Reynolds-number (LRN) *k*-*ω* turbulence model was utilized to resolve the laminar-to-translational flow regimes [54]. Nanoparticles in the range from 0.5 nm to 1000 nm of diameter were investigated. Based on this range, the Peclet number (*Pe* = *LU*/*D*), which is a quantitative measure of advection over diffusion, ranges from 70 to 2.14 × 10^8^. 

Here *L* (= 2.07 mm) is the inlet hydraulic diameter, *U* (= 0.73–2.92 m/s) is the average inlet velocity, and *D* is the particle diffusivity, which is computed following the Stokes–Einstein equation [46,47,55].

The transport and deposition of inhaled nanoparticles were simulated using a discrete-phase Lagrangian model [56,57] with the near-wall velocity correction [58]. In our previous studies, predicted results using this model matched the corresponding in vitro results to a high degree in human upper airways for both ultrafine [59], fine [60] and coarse [61] particles. The governing equation of the particle dynamics is
(1)$$\frac{dv_{i}}{dt}=\frac{f}{\tau_{p}}\left(u_{i}-v_{i}\right)+g_{i}\left(1-\alpha \right)+f_{i,lift}+f_{i,Brownian}\;\mathrm{and}\;\frac{dx_{i}}{dt}=v_{i}\left(t\right)$$
where $v_{i}$ and $u_{i}$ are the particle and flow velocities, $\tau_{p}$ is the particle relaxation time following [62], $g_{i}$ is the gravity and f is the drag coefficient following Morsi and Alexander [63]. The Brownian force on each nanoparticle follows Equation (2) and is updated in each time-step [57]:(2)$$f_{i,Brownian}=\frac{\varsigma_{i}}{m_{d}}\sqrt{\frac{1}{{\tilde{D}}_{p}}\frac{2k^{2}T^{2}}{\Delta t}}$$
here $\varsigma_{i}$ is a Gaussian-based random number, *m_d_* is the particle mass and Δt is the time-step.

The airflow and nanoparticle dynamics were simulated using ANSYS Fluent (Canonsburg, PA, USA), with the computational mesh being generated using ANSYS ICEM CFD (Canonsburg, PA, USA). Tetrahedral mesh was generated with four layers of prismatic cells in the near-wall region. The height of the cell in the first layer was 15 µm. Figure 2e shows the computational mesh in the olfactory region at three scales: a cross-section, a local region and a near-wall region. 

To validate the numerical method in this study, two steps were conducted in this study. First, a mesh sensitivity analysis was conducted by testing different mesh densities, which started from 1.1 million and increased in size incrementally (i.e., 1.1, 1.8, 2.6, 3.6 and 4.9 million). Grid-independent results were assumed when the change in valency at predefined points fell below 0.5% when the mesh size increased from 3.6 million to 4.9 million. The final mesh adopted in this study had approximately 3.6 million cells. Second, numerically predicted deposition was compared to experimental deposition data in the rabbit [15] at normal breathing for micrometer particles (Figure 2f), as any experimental data of nanoparticles are still not available. The close agreement between measured and predicted deposition fractions for inertia particles here, together with our previous validation for nanoparticles [59], gave confidence to the numerical method used in this study. The variability in deposition fractions was investigated using statistical analysis software Minitab (State College, PA, USA). 

## 3. Results

### 3.1. Multiscale Airflows in the Rabbit Noses 

Inspiratory airflows are shown in Figure 3 in different regions of the left nasal passage. Due to its unique architecture, multiscale flow fields are observed, as indicated by the velocity ranges with varying orders of magnitude in Figure 3a–c. Figure 3a shows the stream traces of inhaled airflow that sweep through the maxilloturbinate and enter the lungs. The nasal valve location is indicated by the maximum velocity (red) slightly downstream of the nostril. Figure 3b shows the airflows that are ventilated into the olfactory region. Interestingly, all of these stream traces are from the dorsal meatus and exit the geometry via the apical part of the trachea. Note the velocity range used in Figure 3b is one order of magnitude lower than that in Figure 3a, indicating that the flow speeds in the olfactory region, as well as the flow ventilation into this region, can be much lower than that of the mean flow.

The flow patterns into the sinus are shown in Figure 3c. Even lower velocity range (three orders of magnitude lower the mean flow) has been adopted for the sinus flows, indicating the negligible ventilation into the sinus. The site of the ostium that connects the olfactory region and the sinus is indicated by the black arrow in Figure 3c. The velocity profiles in the two nostrils can be neither symmetric nor uniform, which depends upon the resistance of each passage (Figure 3d). In both nostrils, low-speed flows occur close to the wall, and in the nostril tip, in particular. By contrast, the high-speed core flows occur about 1 mm away from the nostril base. Interestingly, if airflow enter the nose from this position (1 mm away from the nostril base), a perceivable portion reaches the olfactory region (second panel of Figure 3d). Certain airflow also reaches the vomeronasal organ (red dotted circle in Figure 3d). The cross-sectional velocity contours in the maxillo- and ethmoturbinate are shown in Slices 1 and 2, respectively, in Figure 3e. In both contours, the maximum velocities are observed in the dorsal meatus.

Vortices form predominantly in the vestibule for both of the inhalation rates considered (0.68 L/min and 1.91 L/min), which represent normal breathing and sniffing, respectively (Figure 4). This indicates the pivot role of the vestibule in regulating the resistance of respiratory flows [56]. Compared to 0.68 L/min, intensified vortices are also found at 1.91 L/min in the downstream dorsal meatus due to the increased velocity gradients in the near-wall region (Figure 4a). The vortex intensity decreases in the maxilloturbinate region, presumably due to the reduced speed of the airflow through the labyrinthic turbinate region, whose space expands in the mean flow direction. In general, reduced vortex intensities are observed in the deformed models with enlarged vestibule (Figure 4b).

Flow partitions to the superior maxilloturbinate (as delineated in Figure 5a) were quantified for different inhalation flow rates, which range from 28.5% to 37.0%. This flow will most likely penetrate into the ethmoturbinate (or olfactory) region. For all geometry models considered, increasing the inhalation flow rate increased the flow partition to the superior maxilloturbinate region. Expanding the nostril (N1, N2, N3) or enlarging the vestibule also increased the flow partition; however, no significant difference was observed among the four deformed models (Figure 5a). It is still not clear which exact factors caused this seemingly nonpositive relationship between the nostril enlargement level and the level of flow partition. We speculated that enlarging the nostril not only reduced the resistance to the olfactory region, but also to the respiratory region, and thus did not necessarily led to a high partition to the olfactory region. The sensitivity of the flow partition to the nostril/vestibule deformation was further explored by quantifying the flow resistance in the vestibule relative to that in the entire nasal passage, as shown in Figure 5b. Surprisingly, the flow resistance in the vestibule accounts for more than half of the total nasal resistance in the control case for all inhalation rates considered. Expanding either the nostril opening or vestibule noticeably decreases the vestibular resistance. Particularly, the vestibular expansion herein reduces the resistance by 6–8% (Figure 5b). It is, therefore, expected that the geometry changes in the anterior nose during sniffing could perceivably alter the olfactory uptake of inhaled aerosols. 

### 3.2. Spiral Vestibule and Particle Dynamics 

The effect of the spiral-shaped vestibule on inhaled airflow and particles in the nasal passage, and particularly in the anterior nose, were examined using flow visualization techniques (Figure 6). The unique morphology of the vestibule was revisited by displaying it from the top view to emphasize the channels that ramify from the comma-shaped nostril (Figure 6a). Interestingly enough, the tip of the nostril appears to transit to the inferior part of the maxilloturbinate (red line in Figure 6b). Considering the base of the nostril, its lower point connects to the inferior part of the maxilloturbinate (blue line in Figure 6b), while its apical point connects to the top of the maxilloturbinate (black dashed line in Figure 6b). The inhaled airflows in the vestibule are visualized using the snapshots of massless particles that are released into the nostril at varying instants.

The spiral shape of the aerosol cloud is obvious, as the particles travel through the vestibule and twist from the approximately horizontal slit-inlet into the vertical nasal valve (Figure 6c). We also observed that during expiration, particles exited the nose only through the center of the nostril (Figure 6d) because of the spiral, funnel-shaped vestibule and the thin, slit-shaped nostril. Nearly no particle exits through the proximity of the nostril boundary. Particle dynamics in the maxilloturbinate and ethmoturbinate are visualized in Figure 6e in terms of instantaneous snapshots of aerosol positions at varying times after their inhalation into the nose. In both of the inhalation rates considered, particles passing the nasal valve quickly spread into the maxilloturbinate space and are guided into different regions of the nose, as illustrated by the multiple streaks of particles (green and pink color in both panels of Figure 6e). It is also noted that, due to the multi-scale flow speeds, particles inhaled at 0.68 L/min reach the anterior of the ethmoturbinate after 100 ms of being inhaled, while it takes 200 ms to proceed a short distance in the ethmoturbinate, and another 200 ms to move an even shorter distance. Similar observations were also found in the case of 1.91 L/min, with an exception that particles penetrate much deeper into the ethmoturbinate and appear to be lingering there for an extended duration. This particle retention in the ethmoturbinate may have important implications in olfaction. Even though only a small number of particles enter the ethmoturbinate region, they will stay there for a longer duration, allowing them ample time to contact the olfactory receptors by random motion (diffusion).

To better understand olfaction, it is of high interest to know the region of the nostril through which inhaled particles can reach the olfactory region. To obtain this information, the inverse method was used to trace the olfactory-depositing particles back to their release positions at the nostril, as displayed in Figure 7. Particles that deposit in the maxilloturbinate and exit into the lungs are also obtained. To ensure that the results are statistically significant, a total amount of 100,000 particles have been tracked. For all cases considered, the olfactory-depositing particles are from a specific region of the nostril in both passages, which is close to the nostril base (Figure 7a–d). Considering the velocity effects by comparing Figure 7a vs. Figure 7b, a higher inspiratory rate increases both the area (blue ellipse) and deposition fraction to the olfactory region. Nostril expansion (N3) appears to exert an insignificant impact on the release positions of the olfactory-depositing particles (Figure 7a vs. Figure 7c and Figure 7b vs. Figure 7d). It is also noted that the maxilloturbinate-depositing particles are inhaled through the near-wall region of the nostril (black color), while the particles that escape the nasal filtration and enter the lungs are inhaled through the central region of the nostril (green color).

For verification purposes, the olfactory-depositing particles that have been collected in the case of control at 1.91 L/min (ellipse. Figure 7b) were released again into the nostril and their trajectories were tracked again, as shown in Figure 7e. As expected, the majority of released particles traveled along the top ridge of the nose and found their way back into the olfactory region. However, some particles were found to escape into the trachea, which was not surprising considering the stochastic nature of nanoparticle motions. Similar observations were expected for the other three cases in this study.

### 3.3. Nanoparticle Deposition Fraction 

Figure 8 shows the numerically predicted total and sub-regional deposition fractions (DF) for particle size ranging from 0.5 nm to 1000 nm at four inhalation flow rates. As expected, the total DF is higher at the low end of the particle size and then quickly decreases for larger particles for all flow rates considered. For each flow rate, the total DF varies little for particles ranging from 100 to 1000 nm. It is also observed that the peak total DF is not predicted at 0.5 nm, even though it has the highest diffusivity (Figure 8a–d). This is because of the fact that a fraction of these ultrafine particles escape from the nostrils via strong diffusion (or random motion) before they can be transported downstream by the inspiratory flow. The smaller particle size and lower flow rate, the higher the chance an ultrafine particle exits through the mouth and enters the environment. This explains the peak DF at 3 nm particles for 0.68 L/min (Figure 8a), while at 1 nm particles for 2.72 L/min (Figure 8d). Considering the sub-regional deposition, particles are predominantly depositing in the front nose (vestibule and maxilloturbinate region). Only a very small fraction deposits in the olfactory region. The insert in Figure 8a shows the DF in the right olfactory region (R_OL), left olfactory region (L_OL), and trachea, which is one order of magnitude lower than those in the front nose.

Olfactory doses are further plotted in Figure 9 in terms of particle sizes and flow rates. Figure 9a shows the DFs in both the right and left olfactory regions at 0.68 L/min. In this case, significant deposition occurs only for particles ranging from 5 nm to 50 nm. For ultrafine particles smaller than 5 nm, the high diffusivity makes nearly all particles deposit in the front nose and leaves no particles to enter the olfactory region. While for fine particles in the range of 100–1000 nm, both the diffusivity and inertia are negligible; particles closely follow the inhaled flow and do not reach the airway surface through diffusion. The velocity effects on olfactory doses are shown in Figure 9b,c, in the right and left olfactory regions, respectively. Several observations are noteworthy: first, significant olfactory doses occur only in the range of 3–50 nm for all inhalation rates considered. Second, for a given particle size, the olfactory dose increases with the inhalation flow rate. Third, the olfactory dose slightly increases with the particle size for particles larger than 800 nm, presumably due to the increasing particle inertia, but with a much smaller magnitude than that in the range of 3–50 nm. The variability analysis of the left olfactory dose (L_OL) is shown in Figure 9d. Considering the stochastic nature of nanoparticles, each test case (with a specific geometry, flow rate and particle size) was repeated five times to ensure statistically significant results. The highest variability in the OL doses is found for 5-nm and 10-nm particles, followed by 3-nm, 20-nm and 800–1000 nm particles (Figure 9d). Furthermore, the peak OL dose appears to occur at a smaller particle size for a higher inhalation flow rate, as illustrated in the inset in Figure 9d.

To investigate the effects of model geometry on olfactory dosing, the predicted DFs in the left olfactory region (L_OL) are compared between the control case and the four deformed geometries (Figure 10). Elevated olfactory dosing was predicted in all of the four deformed geometries, with either enlarged left nostril (N1, N2, N3), or expanded vestibule, as shown in Figure 10a,d for 0.68 L/min and 1.91 L/min, respectively. In particular, enlarging the left nostril (control, N1, N2, N3) appeared to consistently enhance the deposition into the left olfactory region. The geometry-associated variability was calculated among the control and four deformed models for the olfactory deposition in both left and right nasal passages. In the left passage (Figure 10b,e), the highest variability is observed for particles in the range of 10–50 nm at 0.68 L/min (Figure 10b) and in the range of 5–20 nm at 1.91 L/min (Figure 10e). In contrast, the corresponding variability is absent in the right olfactory region, where the right nostril and vestibule have not been deformed (Figure 10c,f). The percentage variation in the olfactory dose in the left passage due to the geometry change can be as high as 40% (20 nm in Figure 10b and 5 nm in Figure 10e). 

### 3.4. Nanoparticle Deposition Distribution 

The distributions of nanoparticle deposition for different particle sizes are shown in Figure 11. Due to the high sensitivity of particle diffusivity to particle size, strikingly different deposition patterns are predicted among different particle sizes. For 1-nm aerosols with high diffusivity, deposition occurs only in the anterior nose; no particles are observed in the olfactory region and trachea (Figure 11a). High levels of deposition are observed throughout the nose for 3 nm particles (Figure 11b). The deposition pattern becomes more localized with increasing particle size (Figure 11c–f). High olfactory doses are observed for particles of 3, 5 and 10 nm (Figure 11b–d), with significantly reduced olfactory doses for 20 nm particles and larger (Figure 11e,f).

Considering that particles which deposit on the airway surface can overlap with each other and may not accurately represent the local deposition, the surface deposition in Figure 11 is replotted in Figure 12 in terms of the deposition enhancement factor (DEF), which is the ratio of the local deposition to the averaged deposition. Similar to Figure 11a, elevated local deposition of 1-nm particles occurs in the anterior nose, and particularly, in the vestibule (Figure 12a). 

The highest deposition is found for 3-nm particles, with a heterogeneous pattern in the maxilloturbinate region (Figure 12b). As the particle size increases, the intensity of the DEF decreases (Figure 12c–f), indicating the important influence of particle diffusion versus advection on the deposition mechanism. For the inhalation flow rate of 1.91 L/min, the Peclet number (*Pe* = *LU*/*D*) ranges from 196 for 0.5 nm particles to 1.5 × 10^8^ for 1000 nm particles. For particles larger than 20 nm (*Pe* ≥ 3.0 × 10^5^), advection is predominant, and particles do not have enough time to reach the airway wall by diffusion from the core flow. It is noted that, different from Figure 11, the DEF distributions in the olfactory region cannot be displayed in Figure 12 due to the blockage by the sinus.

## 4. Discussion

The knowledge of the anatomy and physiology of the rabbit nose is essential to understand its functions in ventilation modulation, aerosol filtration and olfaction. The ability to filter out inhaled matters is directly related to the defense against infectious diseases [64,65]. The maxilloturbinate region has a very high level of complexity, connecting the upstream vestibule to the downstream ethmoturbinate and trachea. The maxilloturbinate tissue in its cross-sectional view resembles the shape of a reef coral (Figure 3e), which effectively fills up the cavity of the nose while maximizing the surface area. This highly complex, fractal-like structure acts as a scrubber to filter out inhaled aerosol particles and chemicals. To better understand the internal structures of the maxilloturbinate, the cover of the upper maxilloturbinate region was removed in Figure 1d. There are multiple ridges ramifying from the vestibule. Each ridge further bifurcates into 3–5 secondary ridges and then converges at the caudal of the maxilloturbinate region (Figure 1d). The spaces between these major or secondary ridges form narrow conduits for the inhaled flow and aerosols, as illustrated in Figure 4 and Figure 6, respectively. This exquisite, labyrinthine structure provides a highly effective system to remove inhaled infectious agents.

Due to its small cross-sectional area, the resistance in the vestibule accounts for 45–55% of the resistance in the entire nasal passages. As such, variations in the shape and size of the vestibule and valve can substantially alter the inhaled airflow and their partition into different regions. In this study, we examined four deformed nose models in addition to the control case, by expanding either the nostril or vestibule in the left passage. It was found that the flow partition to the superior meatus of the maxilloturbinate increased 4–5% in all cases of the four deformed models (Figure 5a). Besides, large discrepancies exhibit in the coherent vortex structures between the control and the models with an enlarged vestibule (Figure 4). Compared to the control case, the intensity of the vortices decreased in the vestibule and maxilloturbinate region for the deformed models considered.

Besides the flow-limiting feature of the vestibule, another salient feature of the rabbit vestibule is the multiple channels and their spiral morphology. The airspace in the nose atrium transits from the nostril slit into at least five channels (Figure 1d), each leading to different parts of the nose (i.e., superior, middle and inferior). For instance, airflows that are inhaled via the tip of the nostril are predominately ventilated into the inferior maxilloturbinate, airflows via the base of the nostril are mostly ventilated into the upper maxilloturbinate, while only airflows inhaled via a specific area in the nostril can enter the olfactory region (Figure 7). The redistribution of inhaled airflows is mainly facilitated by these spiral channels. In particular, the airflow that can reach the olfactory region can only be inhaled from a specific area close to the nostril base. This observation is true for both the control and deformed models and in both the left and right nostrils. Similarly, expanding the nostril and vestibule also increases the aerosol deposition in the olfactory region (Figure 9 and Figure 10).

It is well known that animals like rabbits have far superior olfactory acuity than human beings, but why? One explanation is that that there are much more olfactory receptors in rabbits (100 million) than humans (6 million) [66], and that like dogs, the part of the brain that is devoted to analyzing smells in rabbits can be proportionally much larger than in humans [24]. Another explanation is that chemicals that travel along the bottom of the nasal passage may be sensed by the vomeronasal organ (red dashed ellipse, Figure 3d), which is also an organ of chemoreception. It is also suggested that the rabbit olfactory mucosa occupies a larger portion of the total nasal airway than that of the human and thus can receive more deposition. However, results herein showed that the deposition fraction of inhaled particles in the olfactory region (less than 1.2%, Figure 9) is lower than its area portion (15.4%) [67]. There must exist some other mechanisms that contribute to the rabbit’s keen sense of smell, and the unique shape of the nostril and the vestibule may be one of them. The rabbit nostril has a curved slit shape due to the obstruction of the naris fold and looks like a comma. The rabbit vestibule looks like a funnel or trumpet and decreases its cross-sectional area in a short distance from the nostril to the nasal valve. Besides, the vestibule spirals from the nostril to the nasal valve and is split into several channels that lead to different parts of the nose. During exhalation, aerosols travel through the funnel-shaped vestibule and exit the nostril only through the middle region of the nostril slit, which leaves the inhaled olfactory-depositing aerosols unmixed with the inhaled ones. This is very different from humans, where exhaled air and aerosols exit through the entire nostril. The anatomical structures underlying this feature are the funnel-shaped vestibule and comma-shaped nostril slit, where the drastic area expansion in the anterior nose does not allow enough time for exhaled particles to fill up the nostril. For the same reason, negative pressure gradient and reverse flows are also possible in both nostrils. Craven and coworkers [24,68] also suggested that the intake of chemicals to the olfactory region in dogs can be continuous during the entire respiration cycle, as opposed to the intermittent olfactory intake, which occurs during the inhalation only. It is noted that continuous flows into the olfactory region were not observed in the steady flows in this study, despite the observation of local reverse flows into the nostril during the inhalation. Transient flows during normal breathing and sniffing should be examined to clarify whether meaningful olfactory intake exists during exhalation under various respiration frequencies and with dynamic vestibule expansion and contraction.

There are multiple assumptions in this study that may affect the physical realism of the results hereof. These include steady flows, rigid airway walls, and one original rabbit nose model. The normal respiration rate of an adult rabbit is 30–60 breaths per minute, while the respiration rate can be 480 breaths per minute during sniffing [67,69]. The transient effects due to the rhythmical inhalations during sniffing can become significant and may result in different flow patterns from those of steady inhalations [70]. The rhythmic movement of the rabbit nose (wiggling, sniffing, etc.) is a natural pheromone, and can have biological implications in regulating the flow ventilation under different breathing modes [61,71]. Moreover, the models considered in this study were reconstructed from images of one single rabbit, which cannot account for intersubjective variability, or represent the population mean [72,73]. A large cohort of rabbit nose models will help to verify the observations in this individual model.

## 5. Conclusions

In Summary, airflow and nanoparticle deposition in an NZW rabbit model were numerically investigated under both normal and sniff-representative conditions. Specific findings include:
Particles that reach the olfactory region come from a specific area in the nostril.The spiral multi-channel vestibule plays an essential role in regulating the partition of inhaled airflow into different parts of the nose.Enhanced olfactory deposition fractions were predicted by expanding either the nostril or the vestibule. Particles in the range of 5–50 nm are more sensitive to the geometry variation than other nanoparticles. The percentage variation in the olfactory dose in the left passage due to the geometry change can be as high as 40%.Exhaled aerosols occupy only the middle region of the nostril, which minimize the mixing with the aerosols close to the nostril boundary, allowing an undisruptive sampling of odorants.The results of this study will have implications in the study of the olfaction in rabbits and of inhalation dosimetry of inhaled submicron infectious agents.


## Figures and Tables

**Figure 1 animals-09-01107-f001:**
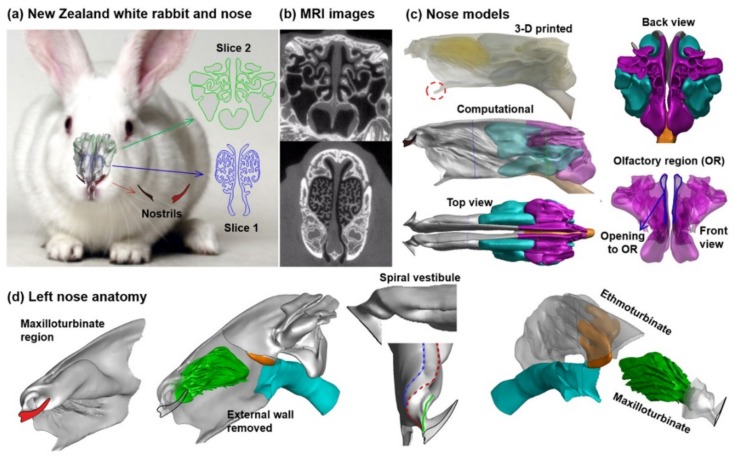
Nose of a New Zealand white (NZW) rabbit: (**a**) the nose model superimposed on a NZW rabbit, (**b**) Magnetic resonance imaging (MRI) scans of the maxilloturbinate and ethmoidal region, (**c**) 3D-printed nose model and computational model separated into three sections: (gray: maxilloturbinate; light blue: sinus; pink: olfactory region) in different views, with the red dotted circle denoting the vomeronasal organ, and (**d**) an assembly diagram of the left nose anatomy: the maxilloturbinate region with and without external walls, the spiral vestibule and the olfactory region (ethmoturbinate).

**Figure 2 animals-09-01107-f002:**
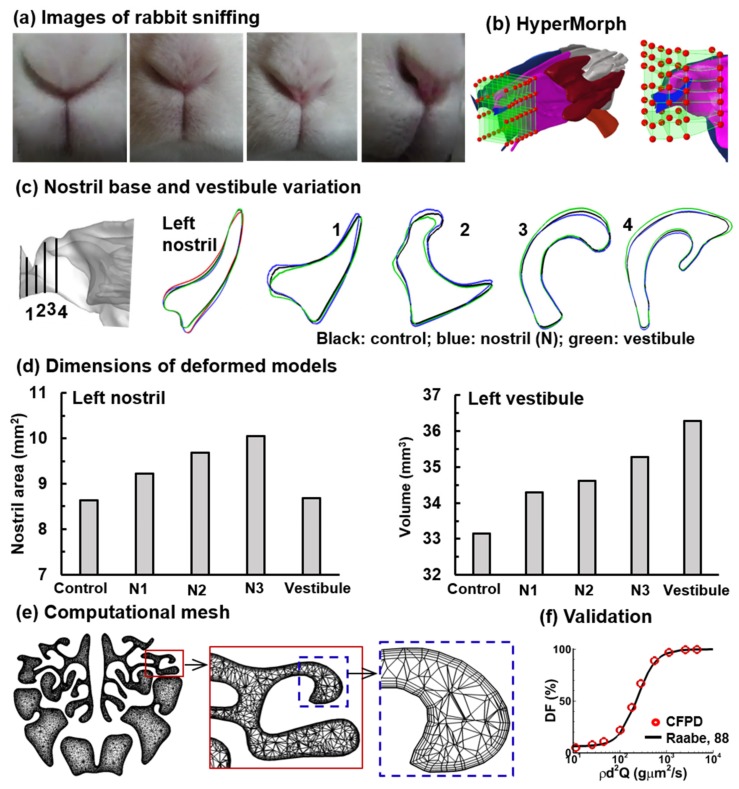
Image-driven deformed nose models: (**a**) video images of the nostrils of a sniffing rabbit, (**b**) variation of the nostril tip, (**c**) models with an enlarged nostril or vestibule, (**d**) dimension of the left nostril and left vestibule in the deformed models, (**e**) computational mesh at three scales, and (**f**) validation with experiments (Raabe, 1998).

**Figure 3 animals-09-01107-f003:**
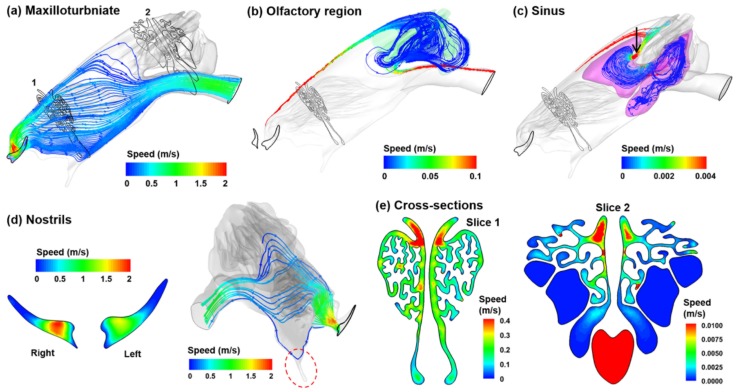
Multiscale airflow velocities in the nose model (left passage): (**a**) maxilloturbinate, (**b**) olfactory region, (**c**) sinus, (**d**) nostrils and (**e**) cross-sectional views.

**Figure 4 animals-09-01107-f004:**
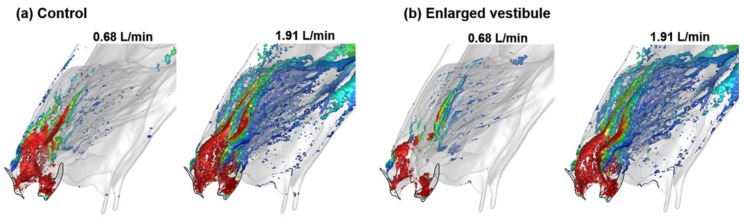
Coherent vortex structures in the rabbit nose at two inhalation flow rates (0.68 L/min and 1.91 L/min): (**a**) control, (**b**) deformed model with enlarged vestibule.

**Figure 5 animals-09-01107-f005:**
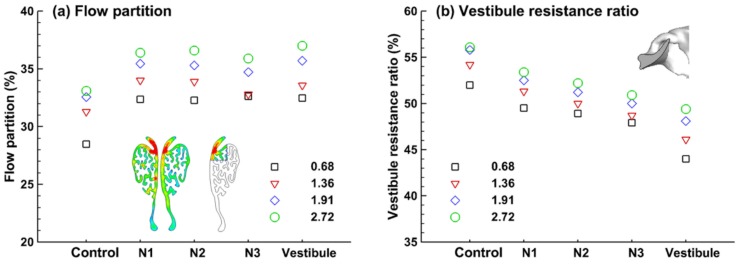
Flow partition and resistance in the left passage: (**a**) flow partition to the superior maxilloturbinate at four inhalation flow rates, and (**b**) the ratio of flow resistance in the vestibule to that in the entire nasal passage at four inhalation flow rates.

**Figure 6 animals-09-01107-f006:**
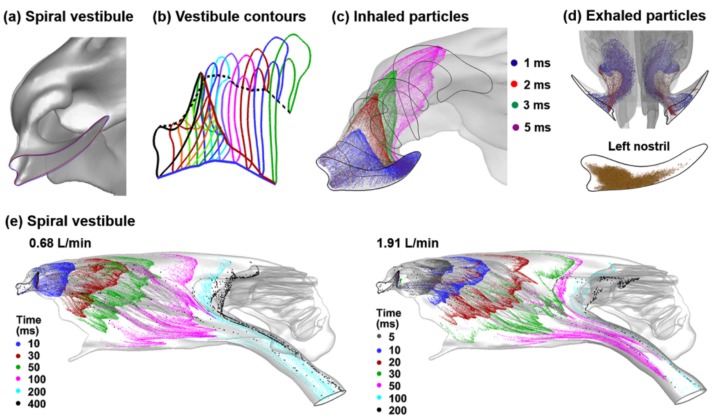
The spiral vestibule and particle dynamics: (**a**) surface model, (**b**) axial contours of the vestibule, (**c**) snapshots of the inhaled particles at varying instants, (**d**) exhaled particles that exit only through the central part of the nostril, and (**e**) the snapshots of particle positions at varying instants at the inhalation flow rate of 0.68 and 1.91 L/min.

**Figure 7 animals-09-01107-f007:**
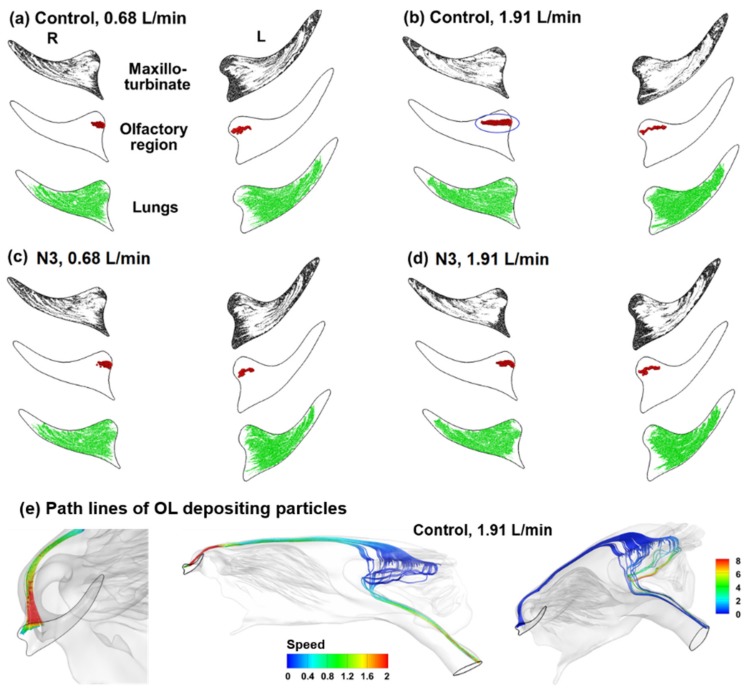
The initial release positions of particles that deposit in either maxilloturbinate, olfactory region, or exit into the lungs: (**a**) control case at the inhalation flow rate of 0.68 L/min, (**b**) control case at 1.91 L/min, (**c**) expanded nostril (N3) at 0.68 L/min, and (**d**) expanded nostril (N3) at 1.91 L/min. The path lines of the olfactory (OL)-depositing particles in the control case at 1.91 L/min are shown in (**e**).

**Figure 8 animals-09-01107-f008:**
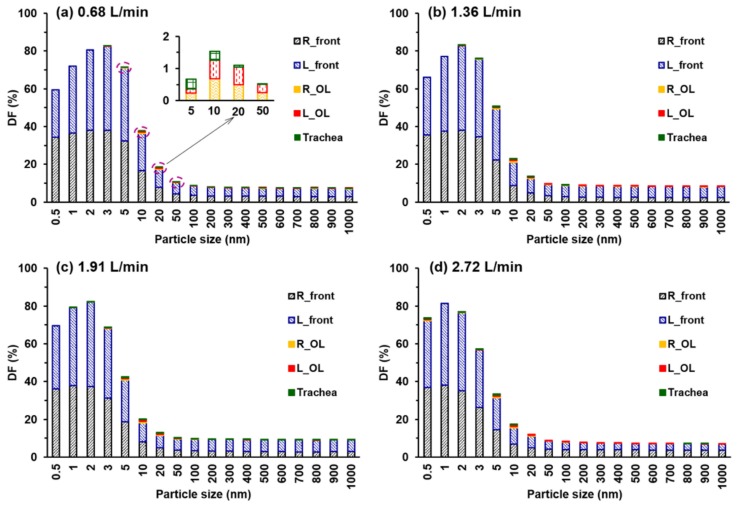
Numerically predicted total and sub-regional deposition fraction (DF) as a function of particle size at different inhalation flow rates: (**a**) 0.68 L/min, (**b**) 1.36 L/min, (**c**) 1.91 L/min, and (**d**) 2.72 L/min.

**Figure 9 animals-09-01107-f009:**
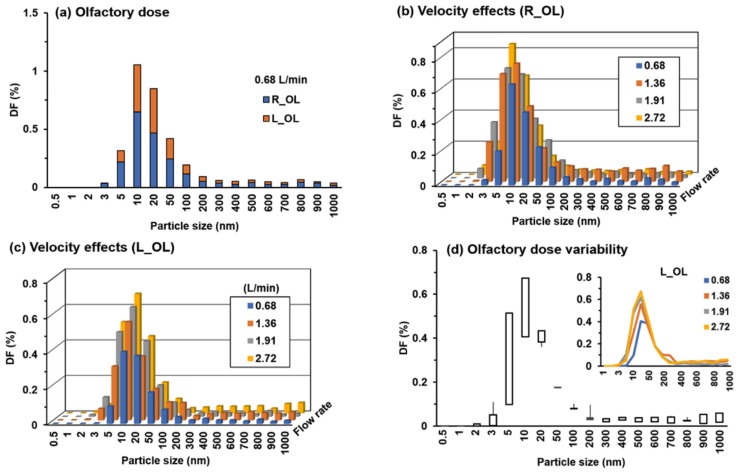
Olfactory doses versus particle size: (**a**) DF in the right and left olfactory regions at 0.68 L/min, (**b**) DF in the right olfactory region (R_OL) at varying inhalation flow rates, (**c**) DF in the left olfactory region (L_OL) at varying inhalation flow rates, and (**d**) variability analysis of DF in the left olfactory region. Due to the stochastic nature of nanoparticles, each test case (with a specific geometry, flow rate and particle size) was repeated five times.

**Figure 10 animals-09-01107-f010:**
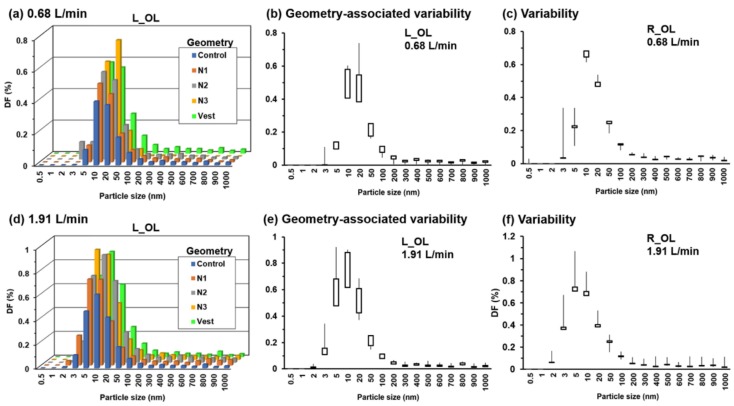
Comparison of the olfactory doses between the control and four deformed cases (i.e., N1, N2, N3, and Vestibule): (**a**) L_OL at 0.68 L/min, (**b**) geometry-associated variability in L_OL at 0.68 L/min, (**c**) variability in R_OL at 0.68 L/min, (**d**) L_OL at 1.91 L/min, (**e**) geometry-associated variability in L_OL at 1.91 L/min, (**f**) variability in R_OL at 1.91 L/min.

**Figure 11 animals-09-01107-f011:**
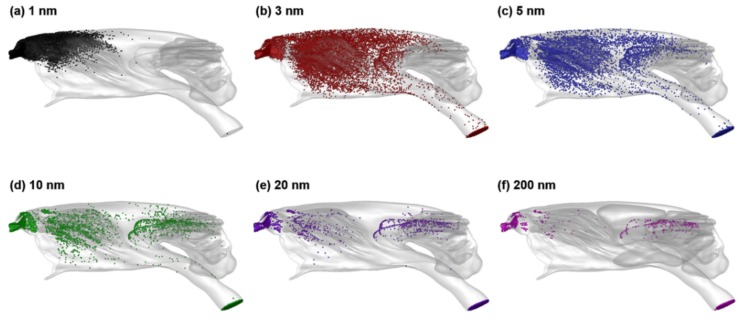
Surface deposition distribution of inhaled nanoparticles at 1.91 L/min for different particle sizes: (**a**) 1 nm, (**b**) 3 nm, (**c**) 5 nm, (**d**) 10 nm, (**e**) 20 nm and (**f**) 200 nm.

**Figure 12 animals-09-01107-f012:**
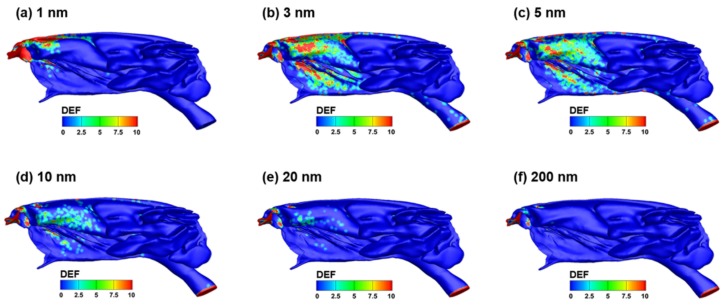
Deposition enhancement factor (DEF) in the rabbit nose at 1.91 L/min for different particle sizes: (**a**) 1 nm, (**b**) 3 nm, (**c**) 5 nm, (**d**) 10 nm, (**e**) 20 nm and (**f**) 200 nm.

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
