# Peer review of "Ventilation Modulation and Nanoparticle Deposition in Respiratory and Olfactory Regions of Rabbit Nose"

_animals, 2019, doi:10.3390/ani9121107_

Round 1

Reviewer 1 Report

In the manuscript “Ventilation Modulation and Nanoparticle Deposition in Respiratory and Olfactory Regions of Rabbit Nose”, authors described the airflow and nanoparticle deposition pattern in rabbit nose using numerical simulation and emphasized that spiral vestibule plays an important role in regulating airflow and olfactory doses. The study results and interpretation are well presented. But there are some concerns need to be addressed.

Some figure captions need to be improved:

Figure 1c. a red dotted circle is shown while it is not mentioned in the text or figure caption; Some for Figure 3d.

Figure 2d. caption is missing

Figure 9. it is written as “Figure 8” in the caption.

According to section 2.3, the size of enlarged nostril in model N1, N2, N3 were determined according to the real condition (video images of rabbit sniffing). How is the vestibule model generated, in other words, how is the level of expansion determined? Figure 5a showed the flow partition to the superior maxilloturbinate in control and all of the four deformed models. Enlargement of nostril and expansion of vestibule all increased the flow partition. But no difference was observed between the four deformed models (e.g., no positive relationship between nostril enlargement level and flow partition level). It would be nice if authors could include a brief discussion about this observation. Typo: line 299, “nave” should be “have” Figure 7c and Figure 7d, which model was used to generate results of expanded nostril? Figure 7e, verification was done in the case of control at 1.91 L/min. Considering the case of control at 1.91 L/min may not be a natural condition (sniffing is accompanied with enlarged nostril), has verification been performed in other cases (e.g., enlarged nostril at 1.91 L/min)? Line 353, “olfactory dose slightly increases with the particle size for particles larger than 800nm”. Is this increase significant, given the variability of particles larger than 800nm is also larger as shown in Figure 9d? Figure 10, how is the Geometry-associated variability calculated? In the conclusions, authors conclude that “expanding the vestibule appears to be more effective in enhancing the olfactory deposition”. However, this conclusion is not obvious from the results. In Figure 10, higher DF was seen in both nostril models and vestibule models depending on the size of particles. It would be good to discuss about more supportive evidence for this statement.

Author Response

In the manuscript “Ventilation Modulation and Nanoparticle Deposition in Respiratory and Olfactory Regions of Rabbit Nose”, authors described the airflow and nanoparticle deposition pattern in rabbit nose using numerical simulation and emphasized that spiral vestibule plays an important role in regulating airflow and olfactory doses. The study results and interpretation are well presented. But there are some concerns need to be addressed.

Some figure captions need to be improved:

Figure 1c. a red dotted circle is shown while it is not mentioned in the text or figure caption; Some for Figure 3d.

Response:

1. The text that mentioned the red dotted circle, which denotes the vomeronasal organ, has been highlighted on lines 145-147.

2. The explanation of the red dotted circle was also added in the caption of Figure 1c, Line 155: “with the red dotted circle being the vomeronasal organ.”

3. Explanation of the red dotted circle in Figure 3d was added in Lines 243-244: “Certain airflow also reaches the vomeronasal organ (red dotted circle in Figure 3d)”.

Figure 2d. caption is missing

Response: The caption of Figure 2d was added: “(d) dimension of the left nostril and left vestibule in the deformed models,” (lines 160-161)

Figure 9. it is written as “Figure 8” in the caption.

Response: The figure caption was changed to “Figure 9.” (line 358)

According to section 2.3, the size of enlarged nostril in model N1, N2, N3 were determined according to the real condition (video images of rabbit sniffing). How is the vestibule model generated, in other words, how is the level of expansion determined?

Response: More details in determining the levels of expansion and generating the deformed models were provided in lines 166-169: “To simulate this change, the widths at the middle of the nostril slit were measured and the maximal width variation was determined. The left nostril was then progressively enlarged to generate three new models N1, N2, and N3 using HyperMorph (Troy, MI), with N3 representing the geometry with the maximal nostril widening.”

Figure 5a showed the flow partition to the superior maxilloturbinate in control and all of the four deformed models. Enlargement of nostril and expansion of vestibule all increased the flow partition. But no difference was observed between the four deformed models (e.g., no positive relationship between nostril enlargement level and flow partition level). It would be nice if authors could include a brief discussion about this observation.

Response: Following the Reviewer’s suggestion, a brief discussion of the observed non-positive relation between the levels of nostril enlargement and olfactory flow partition was provided, as follows.
(lines 261-268): “For all geometry models considered, increasing the inhalation flow rate increased the flow partition to the superior maxilloturbinate region. Expanding the nostril (N1, N2, N3) or enlarging the vestibule also increased the flow partition; however, no significant difference was observed among the four deformed models (Figure 5a). It is still not clear what exact factors caused this seemingly non-positive relationship between the nostril enlargement level and the level of flow partition. We speculated that enlarging the nostril not only reduced the resistance to the olfactory region, but also to the respiratory region, and thus did not necessarily led to a high partition to the olfactory region.”

Typo: line 299, “nave” should be “have”

Response: The typo was corrected. (line 318)

Figure 7c and Figure 7d, which model was used to generate results of expanded nostril?

Response: The model N3 was used to generate the results in Figures 7c and 7d. This information has now been added in both the captions of Figure 7 (line 329) and associated text (line 321).

Figure 7e, verification was done in the case of control at 1.91 L/min. Considering the case of control at 1.91 L/min may not be a natural condition (sniffing is accompanied with enlarged nostril), has verification been performed in other cases (e.g., enlarged nostril at 1.91 L/min)?

Response: We conducted verification for the control case at 1.91 L/min only but believe that similar observations will hold for other cases. One new sentence added in lines 336-337: “Similar observations were expected for the other three cases in this study.”

Line 353, “olfactory dose slightly increases with the particle size for particles larger than 800nm”. Is this increase significant, given the variability of particles larger than 800nm is also larger as shown in Figure 9d?

Response: This increase is statistically significant from 800 nm to 900 nm, and 1,000 nm. However, considering the low deposition rate for 800 nm particles, the magnitude of the deposition increase in this range is very relative to the change in the range of 5-50 nm. The sentence has been modified as:

(lines 372-375): “Third, the olfactory dose slightly increases with the particle size for particles larger than 800 nm, presumably due to the increasing particle inertia, but with a much smaller magnitude than that in the range of 3-50 nm.”

Figure 10, how is the Geometry-associated variability calculated?

Response: The method to calculate the geometry-associated variability was provided in lines 386-391: “The geometry-associated variability was calculated among the control and four deformed models for the olfactory deposition in both left and right nasal passages. In the left passage (Figures 10b and 10e), the highest variability is observed for particles in the range of 10-20 nm at 0.68 L/min (Figure 10b) and in the range of 5-20 nm at 1.91 L/min (Figure 10e). In contrast, the corresponding variability is absent in the right olfactory region, where the right nostril and vestibule haven’t been deformed (Figures 10c and 10f).”

In the conclusions, authors conclude that “expanding the vestibule appears to be more effective in enhancing the olfactory deposition”. However, this conclusion is not obvious from the results. In Figure 10, higher DF was seen in both nostril models and vestibule models depending on the size of particles. It would be good to discuss about more supportive evidence for this statement.

Response: By carefully examining the results, we feel that the statement “expanding the vestibule appears to be more effective in enhancing the olfactory deposition” didn’t have enough supports and has been removed from the manuscript (Abstract, Discussion, and Conclusion). Correspondingly, a new sentence was added in the Abstract and Conclusion, as follows.

1. Abstract, lines 37-38: “Particles in the range of 5-50 nm are more sensitive to the geometry variation than other nanoparticles.”

2. Conclusion: lines 509-511: “Particles in the range of 5-50 nm are more sensitive to the geometry variation than other nanoparticles. The percentage variation in the olfactory dose in the left passage due to the geometry change can be as high as 40%.”

Reviewer 2 Report

Please find the comments below:

Abstract:

The authors need to provide some quantitative metrics for differences in the deposition at the vestibule and olfactory region

Methods: The authors need to include information on the type of mesh used with a picture at the critical parts such as the olfactory boundary wall;

The numerical methods need to include a small paragraph on validation;

The models used thought different simulate are steady conditions with fixed boundary; However, the airway region is continually deforming during sniffing. This can lead to a much bigger impact on the results as compared to other factors.

The authors are encouraged to study the same for moving boundary conditions as an extension of this study.

The conclusion part is fairly good qualitatively;

However, it lacks:

a) quantitative data and;

b) areas where such a study could impact either clinical research on humans or study of toxins

Author Response

Abstract: The authors need to provide some quantitative metrics for differences in the deposition at the vestibule and olfactory region.

Response: Quantitative differences in the deposition at the olfactory region due to the geometry deformation were provided in lines 391-393:

“The percentage variation in the olfactory dose in the left passage due to the geometry change can be as high as 40% (20 nm in Figures 10b and 5nm in 10e).”

Methods: The authors need to include information on the type of mesh used with a picture at the critical parts such as the olfactory boundary wall;

Response: Tetrahedral mesh with four layers of prismatic cells in the near-wall region was used in this study (lines 206-208). A new figure panel (Figure 2e) was added, which shows the mesh at the critical part (olfactory region) at three scales (i.e., a cross-section, a selected local region, and a near-wall region).

The numerical methods need to include a small paragraph on validation;

Response: A new paragraph was added on validation of the numerical method in lines 209-219. In doing so, a new figure panel (Figure 2f) was added that compared experimental and predicted deposition fraction for inertia particles. (Lines 209-219): “To validate the numerical method in this study, two steps were conducted in this study. First, a mesh sensitivity analysis was conducted by testing different mesh densities, which started from 1.1 million and increased in size incrementally (i.e., 1.1, 1.8, 2.6, 3.6, 4.9 million). Grid-independent results were assumed when the change in valency at predefined points fell below 0.5% when mesh size increased from 3.6 million to 4.9 million. The final mesh adopted in this study had approximately 3.6 million cells. Second, numerically predicted deposition was compared to experimental deposition data in rabbit [15] at normal breathing for micrometer particles (Figure 2f), as experimental data of nanoparticles are still not available. The close agreement between measured and predicted deposition fractions for inertia particles here, together with our previous validation for nanoparticles [58], gave confidence to the numerical method used in this study.”

The models used thought different simulate are steady conditions with fixed boundary; However, the airway region is continually deforming during sniffing. This can lead to a much bigger impact on the results as compared to other factors. The authors are encouraged to study the same for moving boundary conditions as an extension of this study.

Response: We fully agree with the Reviewer that the rabbit nose continuously deforms during sniffing, and that some flow features that help the rabbits to achieve their olfactory acuity can only be captured using transient flows with moving boundaries. In this regard, steady conditions with fixed boundaries should only be considered as the initial steps to understand the sniffing.

This limitation and the need for future studies with moving boundaries have been acknowledged in Discussion, lines 491-498 (highlighted).

The conclusion part is fairly good qualitatively;

However, it lacks:

a) quantitative data and;

b) areas where such a study could impact either clinical research on humans or study of toxins

Response:

1. A new sentence that quantitatively described the olfactory dose variation due to the geometry change was added in item 3 (lines 509-511): “Particles in the range of 5-20 nm more sensitive to the geometry variation than other nanoparticles. The percentage variation in the olfactory dose in the left passage due to the geometry change can be as high as 40%.”

2. A new sentence was added in the conclusions (lines 514-515): “5. The results of this study will have implications in the study of the olfaction in rabbits and of inhalation dosimetry of inhaled submicron infectious agents.”

Reviewer 3 Report

Dear Editor,

This is a good study of modeling particle deposition in the olfactory regions of rabbit.

The manuscript is useful to provide researchers information of numerical simulation of nano-particle deposition in the respiratory tract. However, there are still many questions need to be addressed before publication. Below are specific major comments:

1. Authors need to compare and validate the results obtained from numerical simulation with experimental data and other previously published data from the literature.

  2. Please describe in the manuscript that how the authors validate the numerical simulations.    3. Please explain in the manuscript that what the conditions are to allow employing this numerical model.    4. In Figure 3 and 5, it is necessary to provide plot of pressure drop distribution in different regions of the respiratory tract.  

Author Response

This is a good study of modeling particle deposition in the olfactory regions of rabbit.

The manuscript is useful to provide researchers information of numerical simulation of nano-particle deposition in the respiratory tract. However, there are still many questions need to be addressed before publication. Below are specific major comments:

Authors need to compare and validate the results obtained from numerical simulation with experimental data and other previously published data from the literature.

Response: To compare and validate the predicted results with experimental data, a new figure panel (Figure 2f) was added. The associated text was added in lines 214-218: “Second, numerically predicted deposition was compared to experimental deposition data in rabbit [15] at normal breathing for micrometer particles (Figure 2f), as experimental data of nanoparticles are still not available. The close agreement between measured and predicted deposition fractions for inertia particles here, together with our previous validation for nanoparticles [58], gave confidence to the numerical method used in this study.”

Please describe in the manuscript that how the authors validate the numerical simulations.

Response: A new paragraph was added that explained the numerical method validation: (lines 209-218): “To validate the numerical method in this study, two steps were conducted in this study. First, a mesh sensitivity analysis was conducted by testing different mesh densities, which started from 1.1 million and increased in size incrementally (i.e., 1.1, 1.8, 2.6, 3.6, 4.9 million). Grid-independent results were assumed when the change in valency at predefined points fell below 0.5% when mesh size increased from 3.6 million to 4.9 million. The final mesh adopted in this study had approximately 3.6 million cells. Second, numerically predicted deposition was compared to experimental deposition data in rabbit [15] at normal breathing for micrometer particles (Figure 2f), as experimental data of nanoparticles are still not available. The close agreement between measured and predicted deposition fractions for inertia particles here, together with our previous validation for nanoparticles [58], gave confidence to the numerical method used in this study.”

Please explain in the manuscript that what the conditions are to allow employing this numerical model.

Response: Besides offering a better understanding of the olfactory superiority in rabbit (lines 462-490), the results of nanoparticle deposition in the rabbit nose can be applicable in inhalation toxicology and in the study of infectious diseases, as discussed in lines 428-441 (highlighted).

In Figure 3 and 5, it is necessary to provide plot of pressure drop distribution in different regions of the respiratory tract.

Response: Considering that there have already been 12 figures in the revised manuscript, with each figure having multiple panels, a reference (was added in Fig. 3 regarding the information on the pressure drop distribution in the rabbit nose, which used the same geometry and simulated nearly identical inhalational rates (0.68, 1.36, 2.72 L/min).

[55]. Xi, J.; Si, X.A.; Kim, J.; Zhang, Y.; Jacob, R.E.; Kabilan, S.; Corley, R.A., Anatomical details of the rabbit nasal passages and their implications in breathing, air conditioning, and olfaction. Anat. Rec. 2016, 299, (7), 853-868.

Round 2

Reviewer 3 Report

Dear Authors,

Thank you for considering my comments. I accept your paper in the correct format.

Best regards,

Azadeh A.T. Borojeni, PhD